# Research on the Utilization of Saline Alkali Water Resources Based on Two-Phase Flow

**Zhilin Sun, Yang Yang *, Wenrong Tu, Qiuyue Hu and Chaoqun Zhai**

Ocean College, Zhejiang University, Hangzhou 310058, China; oceanszl@163.com (Z.S.);
Twy8718@163.com (W.T.); 13758275011@139.com (Q.H.); chqzhai@126.com (C.Z.)
* Correspondence: yygg1215@163.com; Tel.: +86-188-6800-5100

**Abstract:** In order to reasonably use solar energy to solve problems such as land desertification and soil salinization in Southern Xinjiang, this paper proposes a system combining photothermal and flash evaporation technologies, which use local saline water for desalination treatment to achieve secondary utilization of water resources. Firstly, we introduce the whole system of the photovoltaic desalination plant. As an important heat-collecting element of the system, the solar tube is the key to whether this plant can work efficiently. Then, we carry out the detection and theoretical derivation of data along the tube. We establish a two-phase flow model of saline water in the tube, considering convective heat transfer, and define the formula of the heat collecting efficiency factor. Finally, based on iterative calculation, the temperature trend of the tube and the change law of the two-phase flow are obtained, and the ecological and economic benefits and energy efficiency of the system are analyzed.

**Keywords:** Southern Xinjiang; desalination treatment; two-phase flow; heat collecting efficiency factor; benefits analysis

---

## 1. Introduction

Southern Xinjiang is an area in China that is blessed with abundant solar energy, but has the issue of severe saline alkaline water pollution. As such, it is of great significance to make full use of the available solar energy to solve water pollution problems. The Xinjiang region has the characteristics of long days and sufficient solar radiation. As shown by the annual radiation of the Aksu meteorological station (Figure 1), there is 5467 MJ/m² solar radiation per year on average. (There are many monitoring stations in various places, but the Aksu meteorological station is in the Tarim Basin and located in an area that can best represent the local solar radiation). At the same time, the local solar radiation intensity changes with the seasons as shown in Figure 2.

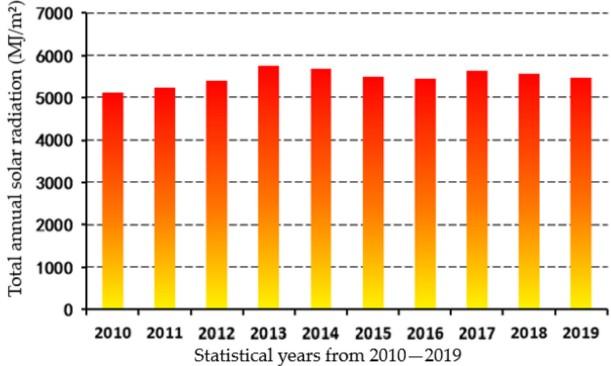

**Figure 1.** Annual solar radiation at the Akesu meteorological station (2010–2019).

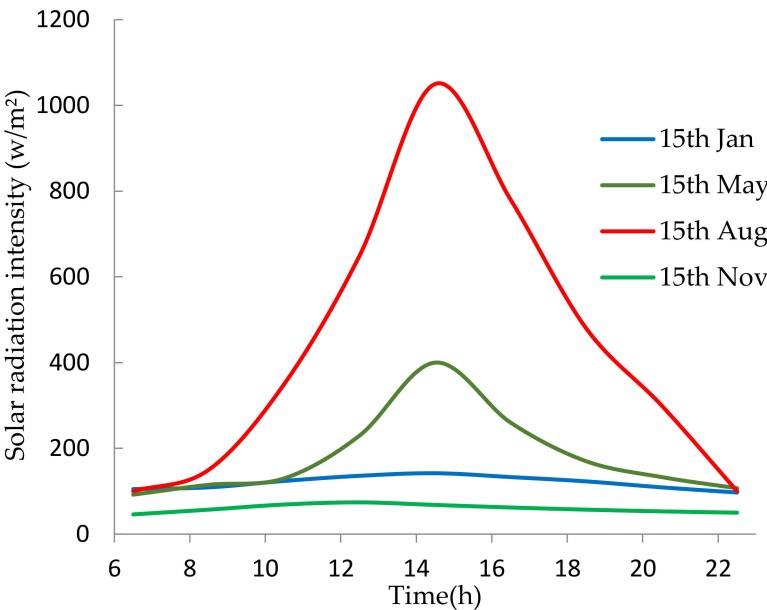

**Figure 2.** Solar radiation intensity in different seasons.

The direct use of solar energy to heat salt water for desalination can meet the growing demand for water resources. Therefore, solar water desalination technologies may become one of the basic water resources for future survival and sustainable development. According to a survey, by 2025, 14% of the global population will be forced to use or drink desalinated water [1]. Solar thermal desalination is a medium-low temperature technology, which poses frontier research problems that have been recognized by the world's new energy and renewable energy departments [2]. Some common desalination devices already exist, such as humidification and dehumidification (HDH), membrane distillation (MD), multistage distillation (MSD), and multieffect evaporation (MED) devices [3–6]. The use of solar energy to achieve the desalination of saline water involves two-phase flow and boiling heat transfer phenomena. Two-phase flow has been widely used in various industrial fields, such as air conditioning [7], humidification [8], water cooling [9], power engineering applications [10], and industrial hot water heating [11]. Furthermore, the number of calculation formulas for flow heat transfer coefficients in the tube has been constantly increasing [12–15]. Badar et al. [16–18] used a steady-state heat transfer model to analyze the performance of single-fluid and dual-fluid pairs of collector tubes under the same conditions.

Thermal distillation in conventional desalination plants often releases large amounts of greenhouse gases due to the consumption of large amounts of energy, leading to global warming [19]. At present, many countries are changing the development direction of seawater desalination technology, focusing on environmentally friendly water treatment processes. Elsarrag et al. [20] studied the advantages of combining solar pools with solar evaporators, recycling the high heat waste brine (brine is a waste product in other desalination systems) as a steam heat source, and using the surface water of the solar pool as cooling water to be used in the condensing system. Their study pointed out that the desalination cost of the solar pond was smaller than that of other desalination methods, and the processing capacity is large. Due to its low efficiency, a solar still cannot be operated on a large scale. The efficiency of solar stills can be sharply increased by adding solar collectors, while most of the energy losses occur as a dissipation of heat, lacking effective energy recycling. Despite the fact that multistage flash distills saline water by flashing a portion of the water into steam in multiple stages, essentially reusing the dissipation energy, its cost performance is still poor.

Mamouri et al. [21–23] developed a salt water desalination device that combines a solar collector tube with an inclination angle of 35 °C and an evaporator, and studied the influence of heat collection temperature and solar radiation on the performance of the collector tube. On this basis, Mosleh et al. [24]

added a trough-type reflector to increase the absorption rate of the collector tube; Li et al. [25] ventilated the bottom of the brine evaporation tank to expand the gas–liquid contact area to increase the amount of water vapor generated. Al-harahsheh et al. [26] added a phase change heat storage tank to improve the salt water desalination device to effectively use heat energy.

These technologies differ greatly in terms of energy consumption, cost, efficiency, and output. The solar distillation process is simple, and the water production efficiency is too low due to the mode of only relying on the natural heating, evaporation, and condensation of the waterbody and the large area. In order to achieve stable working operation, multistage flash evaporation requires additional heat sources, resulting in higher overall energy consumption. The high cost of humidification and dehumidification equipment, the complicated process flow leads to large heat loss, and a relatively high relative humidity is required. The output of water will increase with the increase of humidity. Moreover, the energy transfer of these desalination systems is mostly one-way, and the remaining heat energy after partial heating is not utilized.

Most of Xinjiang is remote with insufficient rainfall throughout the year. Considering the local natural conditions and ensuring a certain amount of water supply, the energy utilization is maximized as much as possible without adding additional heat sources. The system combines the advantages of each process, under the premise of controlling the floor space, input cost, and adapting to the local environment, it achieves a stable water output and provides a method for solving the salt water desalination in remote and arid areas. The whole system mainly includes a water storage tank, condenser, heater, and evaporator. The heater consists of a series of horizontally arranged solar vacuum glass tubes, arranged in parallel. The heat exchanger consists of a condenser, with a heat exchange tube located in the condenser. The cost of the system mainly includes equipment, transportation, construction, and maintenance. The total is about 160,000 USD. The entire system station covers about 400 m$^2$.

Based on the current situation in Xinjiang and the traditional desalination methods, this paper proposes a new desalination system adapted to the local area, described in Section 1. Section 2 introduces the components of the desalination system, and introduces in detail the theoretical analysis method for the core heating element of the system and the establishment of the two-phase flow model. Through field experiments, the temperature changes along the tube were obtained under different flow rates and described in Section 3. The comparison with the efficiency curve of the established two-phase flow model proved the applicability of the model. Through numerical simulation, plots of heat collection efficiency and temperature change under different mass flow rates are drawn. Section 4 explains the relationship between the heat collecting efficiency of the tube and the change of temperature and flow in combination with the analysis of the internal heat transfer mechanism. In the comparison of the advantages and limitations of the traditional desalination process, the practical value of the system is reflected, and the benefits of the system are analyzed from three aspects. Section 5 describes the feasibility of the system for solving the problem of local saline–alkali water desalination, and summarizes the expression of the heat collection efficiency factor and the gas–liquid phase transformation formula established in the theoretical analysis of the two-phase flow of the tube. This article aims to combine the existing desalination technologies to propose a desalination system suitable for Southern Xinjiang, and provide new ideas for solving water resources problems in remote areas.

## 2. Materials and Methods

The benefits of this system are as follows. The heater is made of quartz glass and does not react with saline–alkaline water. Therefore, no more pollution is produced. The glass tube has inner tubes and outer tubes. The inner tube has a heat absorbent coating to absorb most of the solar heat, and the vacuum treatment between the outer and inner tubes can prevent heat loss to the utmost extent. Moreover, the glass tubes are connected in series, which can increase the area for absorbing sunlight and imp heating efficiency. The inside of the evaporator is evacuated, with a flashing function, which

can lower the boiling point of saline–alkaline water and evaporate it into steam in a short time. The hot saline–alkaline water at the outlet of the vacuum glass tube enters the evaporator in the form of a jet stream. The heat exchanger consists of a condenser containing heated steam and heat exchange tubes containing cold saline–alkaline water. The cold saline–alkaline water and steam exchange heat at the surface of the exchange tubes, such that the energy is not easily lost, result from the large heat exchange area. Making full use of the energy of hot steam to heat the cold saline water leads to higher energy efficiency. The schematic diagram of the system as shown in Figure 3.

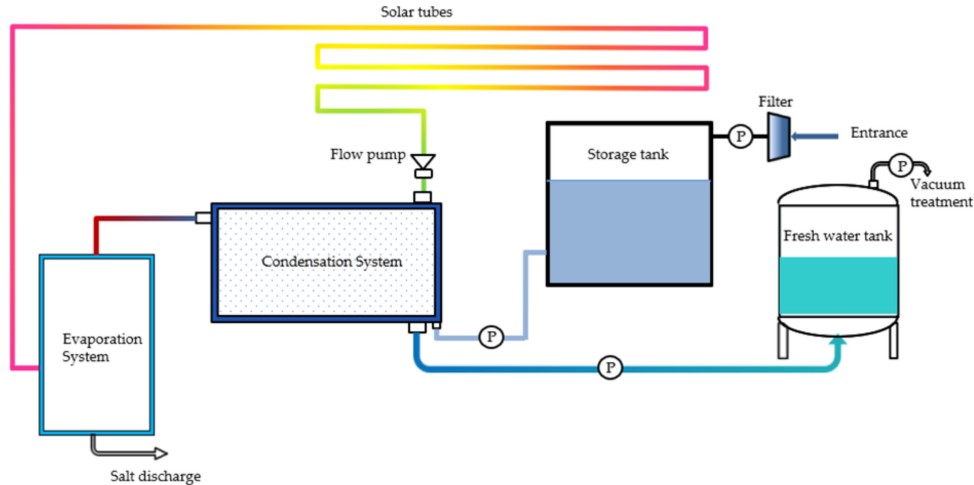

**Figure 3.** Schematic diagram of the proposed system.

Evaporator: The evaporator consists of an evaporation chamber, where the chamber is in a low vacuum state at 0.7–0.8 bar to achieve the purpose of lowering the boiling point of salt water. There is a spin nozzle inside the evaporator, which uses turbulent diffusion to improve the efficiency of gas–liquid separation. The inlet of the evaporator is connected to the outlet of the vacuum glass tube, while the top of the evaporation chamber is connected to the inlet of the condenser. As evaporation continues to occur, fresh water is continuously produced and the salty aqueous solution in the lower part of the flash tank continuously becomes more concentrated, finally reaching the critical salinity (25%) for meeting the salt conditions. At this point, the circulation pump and the return valve are closed, the discharge valve is opened, and the concentrated saline solution in the lower part of the flash tank is discharged. The structure diagram of the evaporator and salt pond are shown in Figure 4.

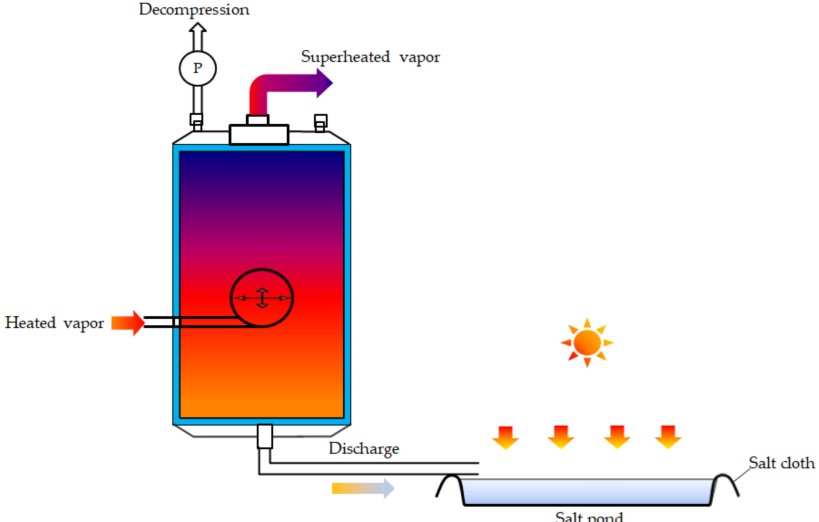

**Figure 4.** Structure diagram of the evaporator and salt pond.

Condensation device: Cold salt water enters the device from the lower part and flows out from the upper part. The cold salt water acts as a continuous cold source for cooling the hot steam into fresh water, while hot steam enters from the upper inlet as a continuous heat source for preheating the cold salt water; finally, the fresh water is collected and removed from the lower outlet. The design of the condenser is shown in Figure 5.

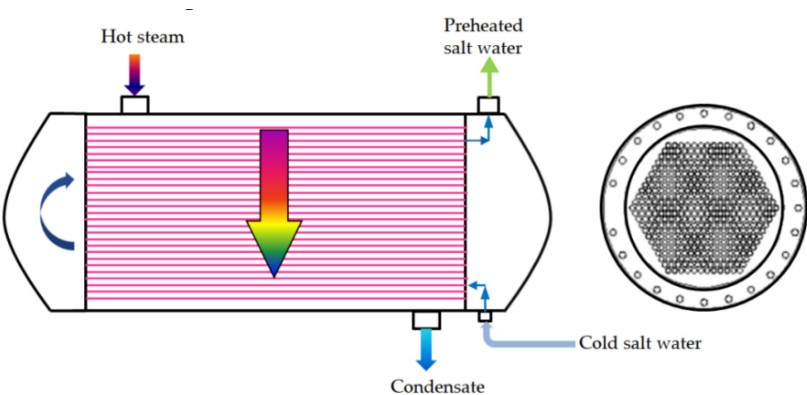

**Figure 5.** Schematic diagram of the condensing preheating system.

Solar vacuum tube: The heat-drying device is made up of many horizontal vacuum tubes, arranged in parallel. The solar vacuum tube used in this study consists of two coaxial quartz tubes, which are divided into an inner tube and an outer tube. The inner tube is coated with a heat absorbing layer, and a vacuum is applied in the gap between the inner and outer tubes. The solar vacuum tube can continuously absorb solar heat and reduce heat loss. By connecting in series, the area of received sunlight can be increased, such that the salty water temperature in the tube increases with distance. The inner and outer tube structure of the solar vacuum tube is shown in Figure 6 and the arrangement of the tubes is shown in Figure 7.

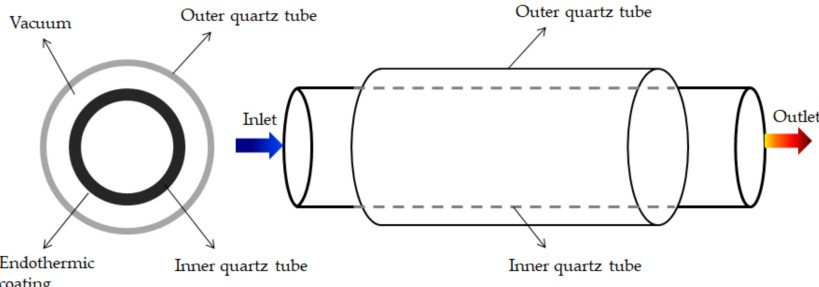

**Figure 6.** Cross-sectional view of a solar vacuum tube and a fluid-flow scheme through the tube.

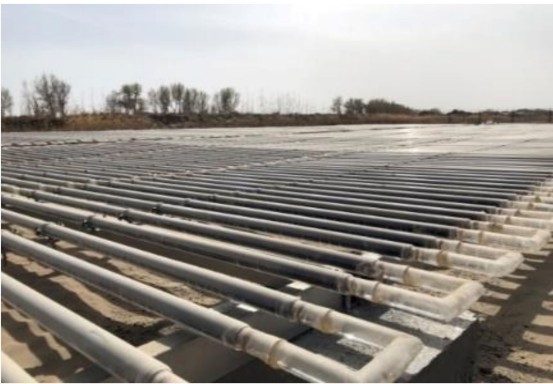

**Figure 7.** Arrangement of the tubes.

### 2.1. Concentrating Heat Mechanism of Tubes

Based on the analysis method of Duffie and Beckman [27], we propose an improved model to meet the requirements of the coaxial effect and two-phase flow research of solar vacuum tubes.

Take a section of tubes for analysis, as shown in Figure 8. The solar radiation energy absorbed per unit area of the quartz inner tube is $S = \alpha\tau \cdot I$, where $I$ is the radiation intensity, $\alpha$ is the absorption rate, and $\tau$ is the transmittance of the quartz outer tube. When salty water flows through the tubes, it absorbs most of the heat, while a small heat loss is recorded as $q_{loss}$. The heat absorbed by the tubes per unit area is:

$$q_{abs} = \alpha\tau \cdot I - q_{loss}. \tag{1}$$

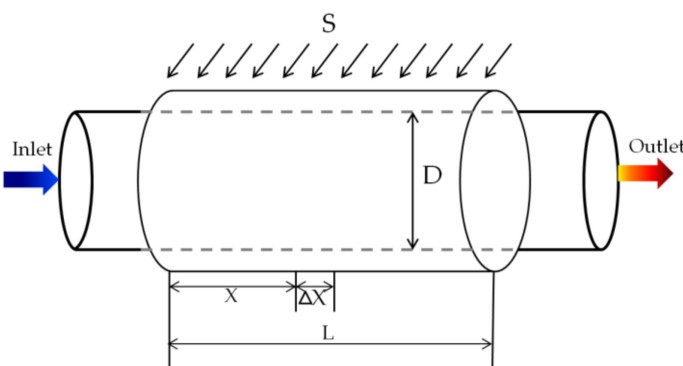

**Figure 8.** Schematic diagram of partial tube section under solar radiation.

The temperature of the endothermic coating gradually rises under the solar radiation, forming a temperature difference with the ambient environment. Heat flows from the endothermic coating through the gas between the inner and outer quartz tubes and then through the outer quartz tube to the surrounding environment. This heat loss is recorded as $q_{loss}$.

In order to calculate the magnitude of the heat loss, the value of the total thermal resistance $U$ needs to be determined. Therefore, assuming that the ambient temperature of the left and right semicircles is the same, the expression of $U$ can be analyzed by considering the tube as an axisymmetric body. The heat loss mechanism of the tube is shown in Figure 9:

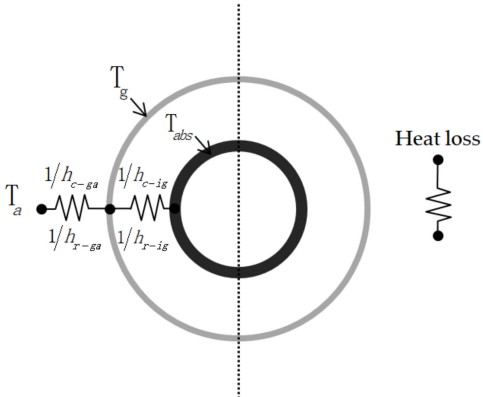

**Figure 9.** Heat loss of a single tube.

This heat loss is caused by radiative heat transfer and convective heat transfer. We obtain the heat loss coefficient of the semicircular tube $U'$ as:

$$U' = \frac{1}{\left(h_{rig} + h_{cig}\right)^{-1} + \left(h_{rga} + h_{cga}\right)^{-1}}, \tag{2}$$

where $h_{rig}$ is the radiation heat transfer coefficient of the endothermic coating of the outer quartz tube (the subscript r indicates radiation), $h_{cig}$ is the sum of the convective heat transfer coefficient and thermal conductivity of the endothermic coating of the outer quartz tube (the subscript $c$ indicates convection and heat conduction, and the subscript $ig$ indicates inner tube to outer tube), $h_{rga}$ is the radiation heat transfer coefficient of the outer quartz tube to the air (the subscript $r$, again, indicates radiation), and $h_{cga}$ is the convective heat transfer coefficient of the outer quartz tube to the air (the subscript $c$, again, indicates convection and heat conduction, and the subscript $ga$ indicates outer tube to the air). When the vacuum between the inner and outer quartz tubes is high enough, convective heat transfer and heat conduction can be isolated (i.e., $h_{cig} \approx 0$). Radiative heat transfer is only considered in the heat transfer process of the inner quartz tube to the outer quartz tube. The radiative heat transfer coefficient between the inner quartz tube and the outer quartz tube is calculated according to the formula of Duffie and Beckman [28]:

$$h_{rig} = \sigma\left(T_g^2 + T_{abs}^2\right)\left(T_g + T_{abs}\right)\left[\frac{1 - \varepsilon_p}{\varepsilon_p} + \frac{1}{F_{ig}} + \frac{\left(1 - \varepsilon_g\right)A_{abs}}{\varepsilon_g A_g}\right]^{-1}, \tag{3}$$

where $\sigma$ is the Boltzmann constant $W/\left(m^2 \cdot K^4\right)$ $\varepsilon_p$ is the emissivity of the endothermic coating, $\varepsilon_g$ is the emissivity of the outer quartz tube, $A_g$ is the surface area of the outer quartz tube, $F_{ig}$ is the angle of view factor of the endothermic coating of the outer quartz tube, $T_a$ is the ambient temperature (the subscript a represents air), $T_{abs}$ is the temperature of the endothermic coating, and $T_g$ is the temperature of the outer quartz tube.

The radiative heat transfer coefficient of the outer quartz tube to the ambient environment is:

$$h_{rga} = \varepsilon_g \sigma\left(T_g^2 + 2T_a^2\right)\left(T_g + T_a\right). \tag{4}$$

The convective heat transfer coefficient of the outer quartz tube to the ambient environment is:

$$h_{cga} = 5.7 + 3.8V_{air}, \tag{5}$$

where $V_{air}$ is the local wind velocity (m/s) and the units of $h_{cga}$ are $W/\left(m^2 \cdot K\right)$.

Under steady-state conditions, the heat transfer per unit area from the endothermic coating to the outer quartz tube is the same as the heat loss per unit area from the endothermic coating to the ambient environment. The net heat increase $Q_{fnet}$ of salty water in the vacuum tube can be expressed as:

$$
\begin{aligned}
Q_{fnet} &= A_{abs}F_c q_{abs} = A_{abs}F_c(\alpha\tau I - q_{loss}) \\
&= A_{abs}F_c[\alpha\tau I - U(T_{abs} - T_a)] \\
&= A_{abs}F_c[\alpha\tau I - 2U'(T_{abs} - T_a)],
\end{aligned} \tag{6}
$$

where $F_c$ is the heat collecting efficiency factor of the tubes, $U$ is the heat loss coefficient of the whole tubes, and $A_{abs}$ is the heat absorbing area of the tube. In the end, we get the water temperature at the outlet of the tube as:

$$T_{fo} = \frac{Q_{fnet}}{\left(\dot{m}C_f\right)} + T_{fi}, \tag{7}$$

where $T_{fi}$ is the water temperature at the inlet, $T_{fo}$ is the water temperature at the outlet (subscript $i$ represents the inlet, subscript $o$ represents the outlet), and $C_f$ is the specific heat capacity of salty water (subscript $f$ represents the waterbody). Salty water enters the inner tubes with mass flow rate $\dot{m}$ (for convenience, hereinafter referred to as the flow rate) and temperature $T_i$, with the temperature of salty water leaving the tube section being $T_o$.

## 2.2. Establishment of the Mathematical Model of Two-Phase Flow in Tubes

Solar radiant energy is continuously absorbed by the tubes, and the waterbody is constantly heated. If the tubes are long enough, the average water temperature at the local position of the tubes will reach the saturation temperature under the corresponding pressure, and flow boiling will occur. In this case, there are two states in the fluid in the tubes, and the ordinary single-phase flow mathematical model is not applicable. It is better to use a two-phase flow model to describe the motion of fluid in the tubes. The heat transfer rate of a two-phase flow is usually much higher than that of a normal single-phase flow. However, a two-phase flow is much more complicated, in terms of physical phenomena, than a single-phase flow, and multiple coefficients need to be considered to predict the heat transfer. It is very difficult to directly study vaporization and heat transfer in the tubes by using a two-phase differential equation. It is, therefore, necessary to introduce the approximate hydrodynamic assumptions:

(1)　The heat flux in the microcell is the same.
(2)　The heat radiation characteristics of the endothermic coating are independent of the initial water temperature.
(3)　The influence of incident angle on the absorption of solar radiation by the tubes can be neglected.
(4)　The ambient temperature around the tubes is the same.
(5)　The flow in the tubes is a laminar or smooth turbulent flow, the Nusselt number is constant ($Nu$), and the flow velocity and water temperature distribution are stable.
(6)　Fluctuations in water temperature and pressure in the two-phase flow can be neglected.
(7)　When the water flowing in the tubes reaches the boiling point, the water temperature in the inner tubes is equal to the saturation temperature, and the heat transfer coefficient is equal to the two-phase boiling heat transfer coefficient.

Taking a microbody of salty water in the flow direction for thermal energy balance analysis (as shown in Figure 9), the average temperature is $T_f$. The microbody absorbs heat from the coating and upstream water, and transfers heat downstream. The heat transferred from the endothermic coating to the salt water is calculated based on the heat transfer coefficient, $h_f$, of the coating to the waterbody. According to Fourier's law, the heat passing through the cross-section of the waterbody per unit time is proportional to the rate of temperature change and the cross-sectional area perpendicular to the cross-sectional direction. The direction of heat transfer is opposite to the direction of temperature rise. The heat transfer of water in the flow direction can be expressed as $-k_f \frac{dT_f}{dx}$, where $k_f$ is the thermal conductivity of water. If vaporization occurs during the flow to produce water vapor, the change in thermal energy in the gas phase is represented by the difference in enthalpy between the outlet and the inlet of the microelement. As shown in Figure 10.

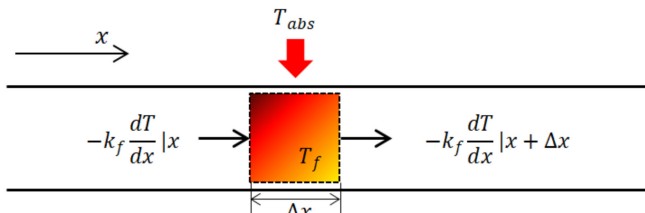

**Figure 10.** Analysis of thermal energy conservation in a microbody of salty water.

The input heat minus the lost heat is equal to the net heat absorption of salty water. According to the heat absorption and heat transfer of the coating, the thermal energy balance equation on the unit coating area is established as:

$$m_{abs}c_{abs}\left(\frac{dT_{abs}}{dt}\right) = \alpha\tau I(t)\pi r\Delta x - \pi D\Delta x q_{loss} - \pi D\Delta x h_f\left(T_{abs} - T_f\right) \tag{8}$$

$$q_{loss} = h_1\left(T_{abs} - T_g\right) = U\left(T_{abs} - T_a\right) \tag{9}$$

where $m_{abs}$ is the mass of the endothermic coating, $c_{abs}$ is the specific heat capacity of the endothermic coating (the subscript *abs* represents the endothermic coating), $h_1$ is the total heat transfer coefficient of the endothermic coating to the outer tubes, $T_g$ is the temperature of the outer tube, $U$ is the total heat transfer coefficient of the heat absorbing coating to the environment, and $T_a$ is the ambient temperature.

When salty water flows through the inner tube, the heat of the tubes is absorbed, the water temperature rises, and the heat energy balance equation of salty water per unit area is:

$$q_{net} = \dot{m}\Delta H \Delta x = \dot{m}c_f\left(\frac{\partial T_f}{\partial x}\right)\Delta x = h_f\left(T_{abs} - T_f\right)\Delta x \tag{10}$$

where $\dot{m}\Delta H$ is the change of enthalpy in salt water.

When the fluid in the tubes is at rest, the constant input of heat causes the water temperature to continuously increase, causing it to be in a nonconstant state. When the water flows in the tubes, the continuously incoming heat is continuously taken away by the water flow, such that the temperature on the local section does not rise with time, showing a near-constant state. The heat absorption of the static fluid in the tubes belongs to the research of time category. If the fluid flows, the heat in the section is taken away, the section temperature is maintained at a stable value, the heat absorbed by the fluid in the next section causes the temperature to be higher than the previous one, and spatial temperature change occurs. This also converts the change of static water temperature with time into the change of flowing water temperature with space, greatly shortening the heating time. As the temperature along the process increases, it is convenient for us to observe the phenomena of vaporization, boiling, heat accumulation, heat transfer, and heat loss in the waterbody, which are also convenient for gas–liquid separation and condensate collection.

### 2.2.1. Heat Collecting Efficiency Factor of Tubes

At both ends of the tubes, there are elbows that are symmetric both upward and downward, with no vacuum insulation. Suppose the two elbows are $2L$ apart, the thickness of quartz inner tubes is $\delta$, and the thermal conductivity is $k_i$. As the endothermic coating is a good conductor, the temperature gradient along the cross-section of the endothermic coating is negligible. As the elbows are exposed to the environment, causing a large heat loss, the temperature of the endothermic coating at the bare place is less than that in the center of the tube section, and part of the heat is transferred from the center to both sides. According to the principle of the heat sink, the temperature at the center of the tubes is the highest, both sides are symmetrically dissipated, and the tubes are symmetric about the center. Therefore, half of the tube is selected for thermodynamic analysis, as shown in Figure 11.

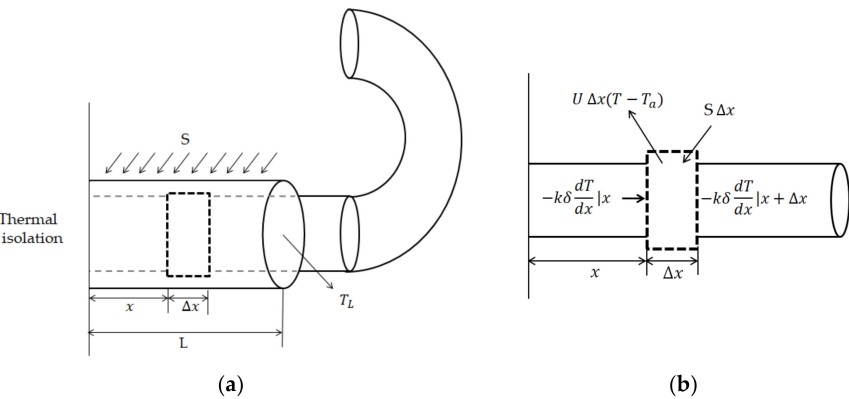

(a)  (b)

**Figure 11.** Analysis of thermal energy conservation in the microbody of the tube; (**a**) is $L$ and the temperature of the endothermic coating at the junction of the straight tubes; (**b**) shows the microbody of the endothermic coating in the flow direction.

The length of the tube section in Figure 11a is $L$ and the temperature of the endothermic coating at the junction of the straight tubes and elbows is recorded as $T_L$. The microbody of the endothermic coating in the flow direction is shown in Figure 11b.

According to Fourier's law, the heat passing through a section in a unit time is proportional to the rate of temperature change, which is perpendicular to the direction of the section. The direction of heat transfer is opposite to that of temperature rise. As a result, heat conduction of the endothermic coating in the flow direction can be expressed as $-k_i\delta\frac{dT}{dx}$, in which $k_i$ is the thermal conductivity of the endothermic coating. The thermal energy conservation equation of the microbody is recorded as:

$$S\Delta x - U\Delta x(T - T_a) + \left(-k_i\delta\frac{dT}{dx}\right)\Big|_x - \left(-k_i\delta\frac{dT}{dx}\right)\Big|_{x+\Delta x} = 0 \qquad (11)$$

By solving this equation, we can define the heat collecting efficiency factor $F_c$ of the endothermic coating as:

$$F_c = 1 - \frac{\tanh mL}{mL} \qquad (12)$$

The tube system is made up of a series of vacuum quartz glass tubes connected in series and the plane layout is in the form of up and down symmetry. Therefore, the heat collecting efficiency factor of each tube section is the same:

$$Q_{abs} = A_{abs}\left(1 - \frac{\tanh mL}{mL}\right)[\alpha\tau I - U(T_{abs} - T_a)] \qquad (13)$$

### 2.2.2. Iterative Calculation of Salt Water Temperature

When the salty water flows through the tubes, the total heat gain of the tube system and the water temperature distribution along the path can be calculated iteratively. We calculate each unit according to the conservation of thermal energy.

The elbow is the n+1 unit, and there is only heat loss at the elbow due to the absence of the endothermic coating and the vacuum insulation. The inlet and outlet water temperatures of unit 1 are $T_{fi1}$ and $T_{fo1}$, the inlet and outlet water temperatures of unit 2 are $T_{fi2}$ and $T_{fo2}$, and so on (i.e., the inlet and outlet water temperatures of unit n + 1 are $T_{fi,n+1}$ and $T_{fo,n+1}$, respectively). As shown in Figure 12.

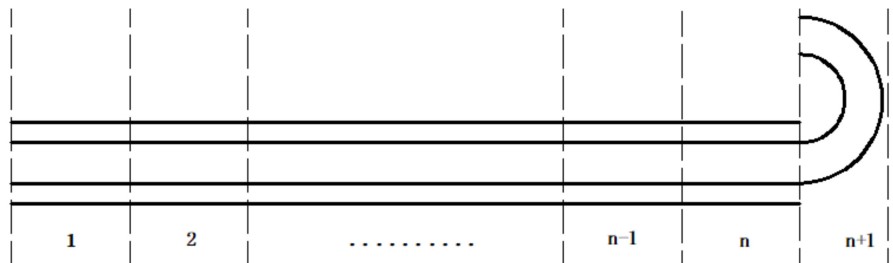

**Figure 12.** Dividing the total length of a tube into n+1 tiny units.

Using Equations (1)–(6), an iterative calculation can be used to obtain the tube's internal temperature distribution:

$$U' = \frac{1}{h_{rig}^{-1} + (h_{rga} + h_{cga})^{-1}}$$
$$= \frac{1}{\frac{(T_g^2 + T_{abs}^2)^{-1}(T_g + T_{abs})^{-1}}{\sigma}\left[\frac{1-\varepsilon_p}{\varepsilon_p} + \frac{1}{F_{ig}} + \frac{(1-\varepsilon_g)A_{abs}}{\varepsilon_g A_g}\right] + [\varepsilon_g\sigma(T_g^2 + 2T_a^2)(T_g + T_a) + (5.7 + 3.8V_{air})]^{-1}} \qquad (14)$$

In the iterative calculation, when the salty water has not reached the boiling temperature, $T_{sat}$, under the pressure of the tubes, it is considered that the tube is a single-phase flow. We use the

single-phase flow heat transfer coefficient $h_f$ to substitute Formula (12) in order to calculate the heat energy absorbed by the unit waterbody. When the unit average water temperature $T_{fm}$ exceeds the saturation temperature $T_{sat}$, the $T_{fm}$ of this unit is set to $T_{sat}$. The net heat increment of gas and water is calculated according to the two-phase flow boiling heat transfer coefficient $h_{f,boil}$. We use the method of Shah [29] to estimate the two-phase heat transfer coefficient:

(1) Calculate the dimensionless parameter $N$:

$$N = C_0 \left(Fr_f > 0.04\right), \tag{15}$$

$$N = 0.38 Fr_f{}^{-0.3} C_0 \left(Fr_f < 0.04\right). \tag{16}$$

(2a) For the dimensionless parameter $N > 1.0$, we calculate the heat transfer coefficients $h_{NcB}$ and $h_c$, according to the following formula (the subscript $NcB$ stands for nucleate boiling, subscript $c$ stands for convection) and choose the larger of the two as $h_{f,boil}$:

$$h_{NcB}/h_f = 230 B_0{}^{0.5} \ (B_0 > 0.0003), \tag{17}$$

$$h_{NcB}/h_f = 1 + 46 B_0{}^{0.5} \ (B_0 < 0.0003), \tag{18}$$

$$h_c/h_f = 1.8/N^{0.8}. \tag{19}$$

The heat flux of the inner tube wall is:

$$\Phi = \frac{Q_{abs}}{\pi D_{in} L}, \tag{20}$$

where $h_f$ is the heat transfer coefficient of the liquid phase, the flow rate of the liquid phase is $G(1 - \alpha)$, and $\alpha$ is the mass fraction of the gas. The heat transfer coefficient of the liquid phase is:

$$h_f = h_{f,in}(1 - \alpha), \tag{21}$$

in which $h_{f,in}$ is obtained from the Nusselt number of a fully developed laminar flow of single-phase fluid in tubes:

$$h_{f,in} = \frac{N_u k_f}{D_{in}}, \tag{22}$$

where $k_f$ is the thermal conductivity of the fluid and $N_u$ is the Nusselt number of the liquid laminar flow.

(2b) For the dimensionless parameter $1.0 > N > 0.1$, the calculation of $h_c$ still uses Formula (21). In the bubble suppression mechanism, it was determined that $h_{NcB}$ depends on the following formula:

$$h_{NcB}/h_f = F B_0^{0.5} exp\left(2.74 N^{-0.1}\right), \tag{23}$$

where $F$ is the efficiency factor of the heat sink. We choose the larger of $h_c$ and $h_{NcB}$ as $h_{f,boil}$.

(2c) For the dimensionless parameter $N < 0.1$, we use Formula (21) to calculate $h_c$ and determine $h_{NcB}$, depending on the following formula:

$$h_{NcB}/h_f = F B_0^{0.5} exp\left(2.74 N^{-0.15}\right), \tag{24}$$

and select the larger of $h_c$ and $h_{NcB}$ as $h_{f,boil}$.

The constant $F$ is determined by the following relationship:

$$B_0 > 0.0011, F = 14.7, \tag{25}$$

$$B_0 < 0.0011, F = 15.4. \tag{26}$$

### 2.2.3. Iterative Calculation of Gas Mass Fraction

One of the main variables of forced convection boiling is the mass fraction $\alpha$ of steam, or vaporization amount, which is used to calculate the latent heat increment, $Q_L$, of the fluid between the two positions in the tube:

$$Q_L = \dot{m}(H_2 - H_1) = \dot{m}\left[\left(H_f + \alpha_2 H_{fg}\right) - \left(H_f + \alpha_1 H_{fg}\right)\right] = \dot{m}H_{fg}(\alpha_2 - \alpha_1) \tag{27}$$

where the subscript L represents latent heat; $H_1$ and $H_2$ are the enthalpies at positions 1 and 2, respectively; and $\alpha_1$ and $\alpha_2$ are the mass fractions of vaporization at positions 1 and 2, respectively.

The steam mass of each unit section can be calculated using Equation (27). Assuming that the water in a unit in the tube reaches or approaches the saturation temperature, the mass of steam at the inlet of unit j is zero. After the saturated waterbody further absorbs the heat of the tubes, steam is generated at the outlet of unit j. The mass of steam flowing from unit j is equal to that flowing into j+1 unit, as shown in Figure 13.

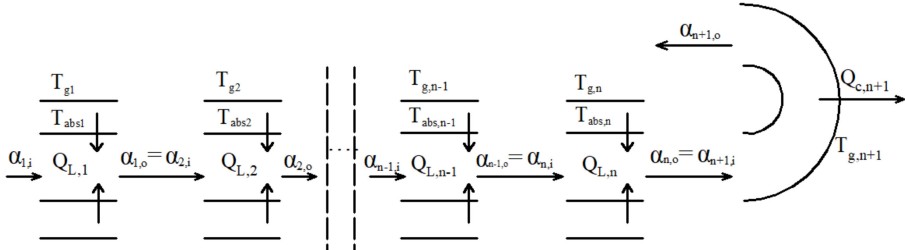

**Figure 13.** Two-phase flow, gas mass fraction of the tube microelement.

For any straight tube unit, the steam mass fraction is:

$$\alpha_{j,o} = \alpha_{j+1,i} = \alpha_{j,i} + \frac{Q_{L,j}}{\dot{m}h_{fg}} \tag{28}$$

For units at elbows, the steam mass fraction is:

$$\alpha_{n+1,o} = \alpha_{n+1,i} - \frac{Q_{c,n+1}}{\dot{m}h_{fg}} \tag{29}$$

where $Q_{c,n+1}$ is the heat released at unit n+1 and the subscript $c$ represents condensation.

### 2.3. Calculation of Theoretical Efficiency of a Single Tube

When using the iterative method to calculate the temperature of the endothermic coating and the heat absorbed by the tubes, we assume that the inlet water temperature of the first unit is $T_{fi1}$, the temperature of the quartz outer tube is $T_{g1}$, and the temperature of the endothermic coating is iteratively calculated as $T_{abs1}$. Then, we determine the value of the total heat transfer coefficient $U$, and then calculate the net heat gain $Q_{fnet1}$ and the outlet water temperature $T_{fo1}$ of this unit. The outlet water temperature $T_{fo1}$ of the first unit is equal to the inlet water temperature $T_{fi2}$ of the second unit, and the process is repeated continuously until the inlet and outlet water temperatures of the $n$th unit are obtained. The heat gain at the elbow unit $n+1$ is negative, satisfying $T_{fo,n} = T_{fi,n+1}$. The average water temperature in the tubes is approximately taken as the temperature of the endothermic coating, and the total heat loss at unit $n+1$ is obtained as:

$$Q_{loss,n+1} = A_{abs}\left[U_{n+1}\left(T_{fm,n+1} - T_a\right)\right] \tag{30}$$

where $U_{n+1}$ is the total heat loss coefficient at unit $n+1$ and $T_{fm,n+1}$ is the average water temperature at unit $n+1$.

The theoretical efficiency $\eta$ of a single solar vacuum tube system can be determined as follows:

$$\eta = \frac{\left(\sum_{i=1}^{n} Q_{fnet,i}\right) - Q_{loss,n+1}}{A_{abs}I} = \frac{\dot{m}C_f\left(T_{fo} - T_{fi}\right)}{A_{abs}I} \tag{31}$$

where $T_{fo}$ and $T_{fi}$ are the outlet water temperature and inlet water temperature of the entire tube, respectively. The relating formula derivation is in the supplementary materials.

### 2.4. Analysis of Two-Phase Flow Simulation

#### 2.4.1. Analysis of the Final Temperature of the Tube

The tube outlet is connected to the evaporator inlet and continuously provides an external heat source for the evaporation system. The inner wall is insulated from the outside. The salt water enters the evaporator after fully absorbing heat in the pipe and gas–liquid separation occurs. The water vapor is continuously pumped away by the exhaust fan at the top of the evaporator, and the salinity of the salty waterbody increases after the evaporation of water, remaining at the bottom of the evaporation tank. In this system, high-temperature salt water enters the evaporator at different flow rates, and a variable-speed axial fan with a maximum capacity of 273 m³/h is used to extract the water vapor generated in the evaporation tank at different steady-state flow rates.

In order to measure the temperature and humidity, a set of monitoring instruments with two humidity sensors and three temperature sensors with a humidity error range of ±3% and a temperature error range of ±0.3 K, respectively, are placed in the evaporator. The sensors monitor the temperature of the high-saline waterbody remaining at the bottom of the evaporator, the humidity and temperature of the gas–liquid separation site at the water inlet, and the gas temperature at the top of the evaporator.

#### 2.4.2. Numerical Simulation of Two-Phase Flow in the Tube

A two-phase flow model was used to numerically simulate the saltwater warming effect after the solar tube absorbs sunlight and heat. We selected the following parameters as input conditions: solar radiation intensity $I = 1000$ W/m², water temperature at the entrance of the drying tube $T_{fi} = 30$ °C, ambient temperature $T_a = 38$ °C, wind speed $V_{air} = 4$ m/s, and flow rates of $\dot{m}_1 = 8.83$ kg/h, $\dot{m}_2 = 4.42$ kg/h, and $\dot{m}_3 = 2.65$ kg/h. The salt content of the waterbody was 5‰. The parameter values required for other theoretical calculations are shown in Table 1.

**Table 1.** The required geometric parameters and constants for theoretical calculation.

| Parameters | Value |
| --- | --- |
| Outer diameter of outer tube (mm) | 35 |
| Inner diameter of outer tube (mm) | 33 |
| Outer diameter of inner tube (mm) | 25 |
| Inner diameter of inner tube (mm) | 24 |
| Emissivity of tube | 0.88 |
| Emissivity of endothermic coating | 0.05 |
| Product of effective transmittance and absorption ($\alpha\tau$) | 0.88 |
| Boltzmann constant | $5.670367 \times 10^{-8}$ |

## 3. Results

### 3.1. Experimental Verification

In order to verify the accuracy of the established theoretical model of two-phase flow, a flow pump was added in the experiment, and three sets of flow rate conditions were selected under the control of

the pump. Three flow rates of 8.8, 4.4, and 2.6 kg/h were selected for the experiment, the inlet water temperature of the tube was 30 °C, and the effective solar radiation monitored on site was 1000 W/m². There were 11 temperature probes along the drying tube, the radiation intensity and initial water temperature of these three working conditions were consistent with the numerical values selected in the simulation, and the variable between each working condition was only the flow rate. Setting the heat transfer in the tube to a constant state based on the heat transfer equilibrium equation, we iteratively solved for the temperature of the water, and numerically simulated the forced convection boiling after the water in the tube reached the saturation temperature and compared it with the experimental data, as shown in Table 2.

**Table 2.** Comparison of experimental (E) and simulated (S) water temperature along the tube.

| Distance along the Tube (m) | Temperature under $\dot{m}_1$=8.81 kg/h (°C) E (S) | Temperature under $\dot{m}_1$=4.4 kg/h(°C) E (S) | Temperature under $\dot{m}_1$=2.62 kg/h(°C) E (S) |
|---|---|---|---|
| 0 | 30 (30) | 30 (30) | 30 (30) |
| 200 | 33 (32) | 36 (35) | 39 (41) |
| 400 | 37 (38) | 43 (42) | 51 (53) |
| 600 | 42 (44) | 51 (51) | 65 (65) |
| 800 | 46 (47) | 57 (59) | 77 (79) |
| 1000 | 49 (52) | 63 (65) | 86 (89) |
| 1200 | 52 (55) | 71 (73) | 95 (100) |
| 1400 | 55 (59) | 79 (82) | 100 (100) |
| 1600 | 58 (63) | 85 (89) | 100 (100) |
| 1800 | 61 (66) | 92 (95) | 100 (100) |
| 2000 | 64 (71) | 100 (100) | 100 (100) |

When the flow rate is 8.8 kg/h, the theoretical efficiency of single-phase flow deviates significantly from the experimental efficiency as the temperature of the heat-absorbing coating increases. When $(T_{abs} - T_a)/I = 0.115$, the deviation can reach up to 12%. The reason is that the single-phase flow model considers the complete vacuum (P = $10^{-5}$ mb), and the convective heat transfer coefficient was ignored in the calculation. However, it is difficult to achieve a complete vacuum during the experiment. The two-phase flow model can consider partial vacuum, which can avoid the deviation between the single-phase flow model and the measured value. At this time, the theoretical efficiency and the experimental efficiency curve are basically in line with the error within 2%. When the flow rate is 4.4 kg/h, the waterbody reaches the boiling point near the outlet, and a smaller amount of steam begins to be produced in the tube. The deviation between the single-phase flow model and the experimental model becomes larger as the gas mass fraction increases. When the flow rate is 2.6 kg/h, the waterbody reaches the boiling point at 1500 m, and vapor begins to be continuously generated in the tube. The heat collection efficiency curve calculated by the two-phase flow model agrees better with the experiment, indicating that the consideration of partial vacuum is also applicable to the gas–liquid two-phase flow state. When the water reaches a boiling state, the temperature of the heat-absorbing coating is at a high temperature, which causes the heat loss to increase, so that the influence of the vacuum factor on the heat collection efficiency of the drying tube decreases in the proportion of the total influencing factors, and the accuracy of the single-phase flow model is somewhat improved.

The heat collection efficiency of the tube is calculated according to the temperature measured under different flow rates. The comparison between theoretical efficiency and actual efficiency of each working condition is shown in Figure 14.

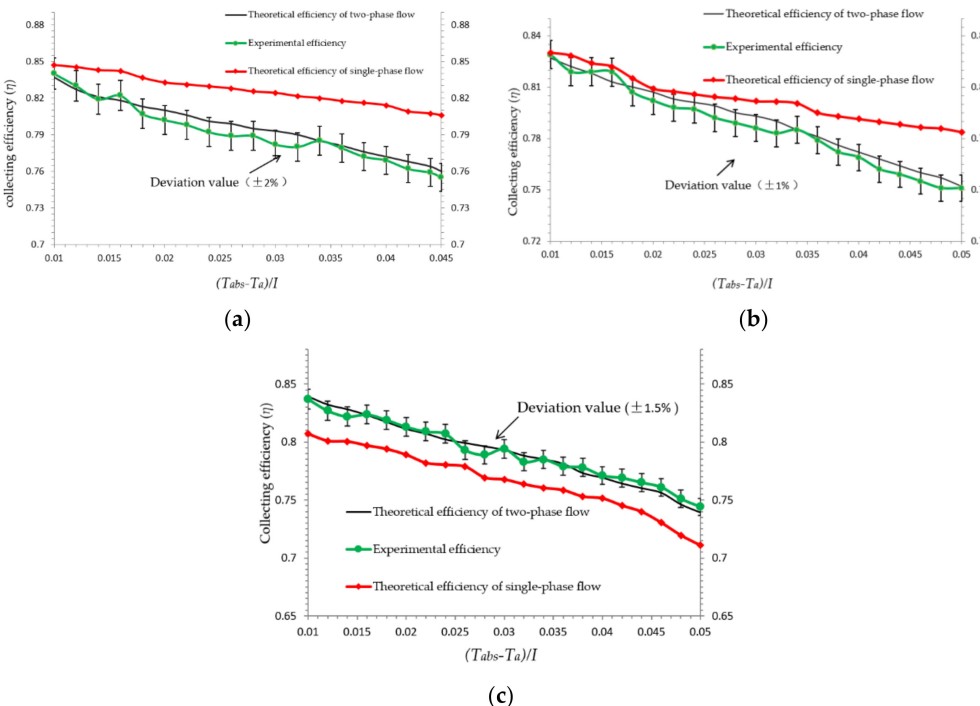

**Figure 14.** Experimental and theoretical efficiency curves with different flow rates; (**a**) Flow rate of 8.8 kg/h; (**b**) Flow rate of 4.4 kg/h; (**c**) Flow rate of 2.6 kg/h.

### 3.2. Analysis of Heat Collecting Efficiency of the Tubes

The outlet temperature of the tubes is mainly related to the amount of solar radiation and the flow rate of the water in the tube. When the amount of solar radiation is constant, the flow of salty water determines the temperature at the outlet. In order to analyze the relationship between the water flow rate and the outlet temperature of the tube, we adjusted the different flow conditions while keeping the solar radiation intensity and initial temperature almost the same as the simulation conditions, and recorded the outlet temperature during steady-state heat transfer under each condition, as shown in Figure 15. When the salty water temperature $T_f$ changed with the flow rate $m$, the correlation coefficient of the trend line reached 0.99 or higher, and the water temperature can be directly calculated by the power function:

$$T_f = 273.79 \dot{m}^{-0.639}. \tag{32}$$

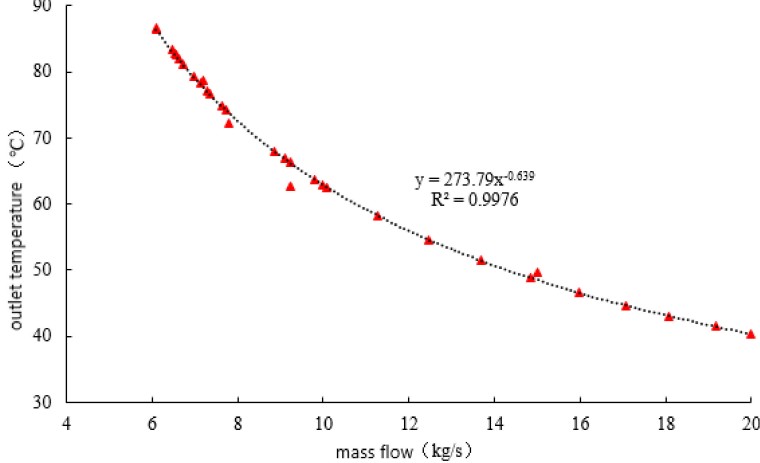

**Figure 15.** Curve of outlet temperature and mass flow rate.

According to the numerical changes obtained by simulation, the relationship between heat collecting efficiency, mass flow rate, and water temperature is established, as shown in Figure 16.

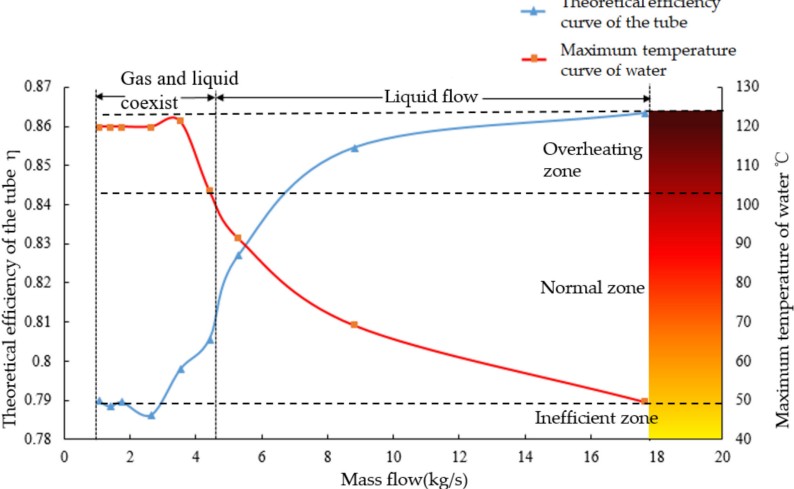

**Figure 16.** Theoretical efficiency of the tube changes with flow rate.

When the maximum water temperature is less than 50 °C, the actual heat transfer efficiency of the tube is low, and called a low efficiency zone. The latent heat of vaporization is generated when the water temperature exceeds 100 °C. The theoretical efficiency of the tube is no longer based solely on the temperature change of the waterbody. Therefore, when the maximum water temperature is higher than 100 °C, it is called an overheating zone.

When the flow rate is reduced from 18 to 4.4 kg/h, the heat collecting efficiency of the liquid flow drops by 6.7%. When the flow rate is lower than 4.4 kg/h, the slower the salt water velocity is, the easier it is to reach the boiling condition, but the heat loss is also greater. In the two-phase flow zone, as the share of the gas phase continues to increase, local deceleration, backflow, and stagnation will occur, so that the water temperature no longer rises but remains at the level of 120 °C; the heat collecting efficiency is reduced to 0.79 nearby, and slightly fluctuates. As the flow continues to decrease, the liquid phase can almost completely vaporize before the water flows out of the tube. In order to prevent accidents due to the excessive share of the gas phase in the tube, it is necessary to set up a pressure relief valve. In addition, after the two-phase flow leaves the tube, the gas will take away the stored latent heat, thereby reducing the heat collecting efficiency. To sum up, with the change of flow, the maximum temperature of the flowing water and the heat collecting efficiency of the tube show the opposite trend. That is, the heat collecting efficiency decreases with the decrease of the flow rate, and the water temperature increases with the decrease of the flow rate. The theoretical analysis of two-phase flow can reasonably describe this phenomenon.

## 4. Discussion

### 4.1. Theoretical Analysis

There is pressurized fluid in the tube, which has a certain gas phase mass fraction. After the salty water enters the evaporation system, the space suddenly increases and the exhaust fan at the top continuously draws air, causing the pressure in the waterbody environment to fall, the boiling point of the waterbody to decrease, and a part of the waterbody to become overheated. As the saturation temperature at the overheated state is lower than the original boiling point, the evaporation rate of the waterbody is accelerated (or even boiling) and the vaporization needs to absorb a large amount of latent heat. As the external heat source of the evaporation system, the tube directly determines the efficiency of gas–liquid separation in the evaporation system. The higher the inlet water temperature, the more obvious the gas–liquid separation effect. The heat transfer in the tube is set to a constant state, such that

the temperature change of the static waterbody with time is transformed into the temperature change of the moving waterbody with space, which is conducive to observing the phenomena of the waterbody starting to vaporize and boil, heat accumulation, heat transfer, and heat loss. Thermodynamic analysis of the phenomenon of heat dissipation from the endothermic coating at the center to the two ends gives the relationship between the heat collection efficiency factor $F_c$ of the tube and the tube length L and the total heat transfer coefficient $U$ of the endothermic coating to the environment. The heat transfer mechanism of the waterbody in the tube is obtained theoretically, and a two-phase flow model on the water temperature and gas phase mass fraction in the tube—that is, the heat and mass transfer balance equations—is used to analyze the critical conditions of water vaporization and boiling, as well as to calculate the thermal efficiency of the tube. Based on the heat transfer equilibrium equation, the temperature of the endothermic coating and the waterbody in the tube can be iteratively solved. Considering that overpressure occurs during the actual operation of the tube, the boiling point of the waterbody becomes larger as the pressure increases during the numerical simulation of the two-phase flow. When the flow rate is reduced from 8.81 to 4.4 kg/h, the heat collection efficiency in the liquid flow zone is reduced by 6.68%. When the flow rate is lower than 4.4 kg/h, such as 2.62 kg/h, the fluid in the tubes is a gas–liquid two-phase flow, indicating that the slower the flow rate of the salt water, the easier it is to reach the boiling condition; however, the heat loss of the tubes is also greater. In the two-phase flow zone, as the proportion of the gas phase increases, the phenomena of deceleration, recirculation, and stagnation will occur in the tubes, such that the water temperature in the drying pipe will not rise and instead remain at 120 °C. Therefore, the heat collection efficiency is reduced to 0.79 and only slightly fluctuates. As the flow rate continues to decrease, the liquid phase is almost completely vaporized before the water flows out of the tubes. In order to prevent accidents caused by an excessive proportion of gas phase in the tubes, it is necessary to install a pressure relief valve. In addition, after the two-phase flow leaves the tubes, the gas will carry away the stored latent heat, thereby reducing the heat collecting efficiency of the tubes. As a result, we introduce this latent heat into the evaporation system, along with the salty water, and release it to drive the evaporation system for gas and liquid two-phase separation.

In summary, as the flow rate changes, the maximum temperature of the flowing water and the heat collection efficiency of the tubes are opposite; that is, the heat collection efficiency of the tubes decreases as the flow rate decreases and the water temperature increases as the flow rate decreases. The theoretical analysis of two-phase flow can reasonably describe this phenomenon.

*4.2. Comparison*

The water production cost of desalination depends on capital cost and operation: equipment and installation, energy, maintenance, and replacement. The energy cost and capital cost occupy nearly 81% [30] of total water production cost of conventional membrane distillation. By comparing the existing conventional desalination treatment, the technologies differ greatly in terms of energy consumption, cost, efficiency, and output due to their different processes. It can be seen from Table 3 that the energy consumption of membrane treatment is far lower than other methods. At the same time, the high cost of the core component of the reverse osmosis membrane and the high frequency of replacement have led to an increase in the overall water production cost. The solar distillation process is simple but has a large area and low efficiency. The output of multistage flash evaporation is low, and the energy consumption is huge due to the need for an external stable heat source. The humidification and dehumidification equipment are expensive, the process flow is complicated, and the heat loss is large. Combining the characteristics of population distribution and water resources in Xinjiang, the system not only controls the overall installation and operating costs, but also ensures that an appropriate amount of water can be produced without chemical treatment, and the local wastewater is reasonably recycled, thereby appropriately solving the local agricultural water shortage and saline–alkali land management problems. The cost of the system mainly includes equipment, transportation, construction, and maintenance. The total is about 160,000 USD.

**Table 3.** Comparison of conventional desalination.

| Process | Core Components | Outcome | Advantages and Limitations |
|---|---|---|---|
| Humidification–dehumidification [31] | Air collector<br>Solar flat plate air heater<br>Humidifier<br>Heat exchanger | Specific energy consumption: 31.1 kWh/m$^3$<br>Plant capacity (m$^3$/day): 1–100<br>Cost of water: 10.5–19.5 USD/m$^3$ | High cost of equipment<br>The system is more flexible<br>Low installation and operation costs<br>Any kind of low grades energy can be utilized<br>Requires large number of stages for efficient operation<br>Water production cost is higher |
| Membrane distillation [32] | Reverse osmosis membrane | Specific energy consumption: 1.2–6.0 kWh/m$^3$<br>Plant capacity: 0.5–29.75 L/m$^2$h<br>Cost of water: 6.5–9.1 USD/m$^3$ | Highly saline feed can be treated<br>Low temperature operation<br>Large membrane surface area is required because of low driving force<br>Membranes are expensive with short life time<br>Biological fouling of membrane is possible |
| Multistage flash [33] | Flash tank<br>Vacuum pump<br>Condenser | Specific energy consumption: <144 kWh/m$^3$<br>Plant capacity (m$^3$/day): 0.009–10<br>Cost of water: 2.6–6.3 USD/m$^3$ | Suitable for large scale production of distilled water<br>The plant is more reliable<br>Plant can tolerate feed water of any quality<br>High quality distillate is produced<br>High energy consumption<br>The plant is heavy and costly |
| Solar still [34,35] | Solar still<br>Solar ponds<br>Flat plate collector | Specific energy consumption: 640 kWh/m$^3$<br>Plant capacity (m$^3$/day): <100<br>Cost of water: 1.3–6.5 USD/m$^3$ | Solar stills can be constructed with locally available materials<br>Minimum maintenance and operation cost<br>Ecofriendly<br>Product water is of high quality<br>Low distillate yield per m$^2$<br>Requires large area<br>Low efficiency |
| New system in this paper | Solar tube<br>Evaporator<br>Heat exchanger | Specific energy consumption: 70–150 kWh/m$^3$<br>Plant capacity (m$^3$/day): <100<br>Cost of water: 3.3–8.7 USD/m$^3$ | Suitable for inland and rural areas<br>Recycle waste saline water<br>Improve agricultural environment<br>Low water production<br>Affected by local sandstorms<br>Water production efficiency of the system is unstable |

*4.3. Benefit Analysis*

Economic benefits: Southern Xinjiang is located in the southern foothills of the Tianshan Mountains, with an area of about 10% of China. The climate is dry and the annual evaporation is 1877.5–2337.4 mm. Due to the generally high soil salinity in the Tarim Basin, the salt content of the 0–30 cm soil layer is 50–300 mg/kg, reaching a maximum of 600–800 mg/kg at the surface, where 47% of the area is abandoned due to secondary salinization; the actual reserved area is only 18,666.7 km$^2$ [36]. According to a survey [37], soil salinization reduces Xinjiang's grain output by about 720 million kg/year, accounting for about 8.6% of the total grain output of the whole year, causing economic losses of about 3.5 billion RMB. For conventional desalination equipment [38,39], the site construction is restricted by many geographical conditions, and it covers a large area and has a high investment cost. The vast area of Xinjiang, the uneven population distribution, and the characteristics of a large number of remote rural areas all need to be taken into account [40]. There are many drainage canals in various regions, combined with the advantages of the system's total area of less than 400 m$^2$ and the low construction cost, it is a good choice to improve local economic development.

Ecological benefits: The survey showed that in the Aksu irrigation area, the current water consumption rate of cotton is 0.65–0.75 kg/m$^3$, and the water consumption rate for wheat is 1.2 kg/m$^3$. This shows that there is still room for improvement in the water productivity of wheat and cotton in this area, while the economic benefits of cotton are better than those of wheat. There are a lot of cotton fields near the system. Due to the greater water consumption of salt field cultivation, the system can improve the water supply by purifying the waste saline water in the drainage canals without sacrificing the original ecological stability. The water is used to grow local cotton fields, effectively improving the local ecological planting environment.

Energy benefits: The energy utilization efficiency of the system is an important parameter to measure its feasibility and benefit. Through the observation records of the inlet and outlet water temperatures, radiation, gasification, and other parameters of the pipe section, the total heat absorption of the heat-absorbing coating is calculated from the heat collection efficiency factor as $F_c$: $Q_{abs} = A_{abs}F_c[\alpha\tau I - U(T_{abs} - T_a)] = 172,251.43$ KJ, compared with the actual heat gain calculated by the temperature change of the inlet and outlet and the latent heat of vaporization: $Q_{net} = Q_f + Q_{fg} = 145,896.97$ KJ, where $Q_f$ is the sensible heat and $Q_{fg}$ is the latent heat of vaporization. The energy utilization efficiency $\eta$ can, then, be obtained as: $\eta = \frac{Q_{net}}{Q_{abs}} \times 100\% = 84.7\%\%$, which reflects the feasibility of the energy efficiency of the tubes.

In the future, mainly the heat loss of the tube will be studied. The heat loss at the elbow is relatively small (compared to the straight pipe). The experimental and numerical simulation results show that the accuracy is high enough when only analyzing the heat loss of the straight. However, if the heat loss of the elbow can be considered in the research, the accuracy will be further improved.

## 5. Conclusions

Conclusion 1: Considering the problems of strong sunshine, dry climate, and shortage of water resources in Xinjiang, this paper combines solar evaporator and flash evaporation technologies to propose a method for the separation of brackish water based on concentrated heat. Taking into account the characteristics of high temperature resistance and corrosion resistance of quartz tubes and evaporators, sufficient daily routines ensure the stable operation of the system, maximize the utilization rate of the system, and minimize the loss rate.

Conclusion 2: In the study of the system, through the monitoring, numerical simulation, and iterative calculation of the tube, the phase transition formula of the gas–liquid two-phase was obtained as $\rho_v c_v \left(\frac{dT_v}{dt}\right) = \rho_f \frac{duH}{dx}$. Through thermodynamic analysis of the phenomenon of heat dissipation of the endothermic coating at the center along the pipe section towards both ends, the relationship between the heat collection efficiency factor $F_c$ in the tube and the length $L$ of the tube was obtained as $F_c = 1 - \frac{\tan mL}{mL}$. The total heat transfer coefficient $U$ of the coating to the environment

could then be obtained, which revealed the heat transfer mechanism of water. This provides a theoretical basis for ensuring the stable and effective operation of the system.

**Supplementary Materials:** The following are available online at http://www.mdpi.com/2073-4441/12/11/2994/s1.

**Author Contributions:** Data curation, C.Z.; Methodology, Y.Y.; Resources, W.T.; Software, Q.H.; Writing—original draft, Y.Y.; Writing—review & editing, Z.S. and Y.Y. All authors have read and agreed to the published version of the manuscript.

**Funding:** This research received no external funding.

**Conflicts of Interest:** The authors declare no conflict of interest.

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
