# Peer review of "Research on the Utilization of Saline Alkali Water Resources Based on Two-Phase Flow"

_water, doi:10.3390/w12112994_

Round 1
Reviewer 1 Report
Dear Authors,
thank you for considering my comments and making corrections to the manuscript. The responses to my comments are clear and to the point, and more importantly they improve the scientific character of the presented work.
Therefore, I accept the responses presented and the changes introduced to the paper and recommend it for publication in Water.
Regards.
Author Response
Dear reviewer:
Thank you very much for your approval of the revision of my manuscript and your help in my research.
Sincerely,
Yang Yang
Reviewer 2 Report
In this article, Zhilin Sun et al discuss a well explained work on a practical desalination system based on two-phase flow. The work is detailed in its depth of study and experimental verification of the described concepts. However, I have a few concerns with regard to novelty of the work and presentation for publication in 'water'.
- In figure 1, the authors present the annual solar irradiation data in MJ/m2. However, standard practice is to describe this parameter in kW/m2 or W/m2. I recommend the use of these standardized units in order to make it easy and relatable to the readers.
- As described by the authors in the first three paragraphs of the introduction, several detailed analytical and experimental studies have been carried out on solar desalination systems consisting of improved solar collectors, evaporators, heat exchangers and condenser systems. The authors state that these conventional systems have drawbacks including "low water production efficiency, waste of condensing heat production, and difficulty in reusing energy." However, the authors do not elaborate on what exactly leads to these issues in previous systems and how specifically this study addresses these challenges. Adding a section on this is critical in my opinion.
- The authors claim that this study addresses challenges associated with providing water resources to remote areas. However, the components used in these systems are similar or more cost intensive compared to conventional systems. Furthermore, it is fairly labor intensive to set up these systems in remote areas. The authors need to justify novelty based on cost benefits in order to justify this statement.
- As shown in figure 1, the salt is discharged from saline in the evaporator. A major issue in such desalination systems is the accumulation (fouling/clogging) of salt over prolonged durations which also affects the efficiency of operation in these systems. The authors need to comment/discuss as to how this issue is handled in these systems and how it plays a role in overall operation.
- The authors need to check the entire manuscript for grammatical errors and spell checks.
Reviewer 3 Report
In this work a system that combines photometric technology and flash evaporation to use solar energy to obtain water for water resources is exposed.
First, a desalination plant based on solar energy is presented. The heat collector is the most important element in the system. It consists of a solar tube where the heating and evaporation process is carried out. An important theoretical development of this collector forms the bulk of the article, describing the model of the flow of saline water in the tube, considering the transfer of heat by convection, the formula for the efficiency factor of heat collection. An additional contribution has been the iterative calculation to obtain the temperature values ​​along the collector tube.
The efficiency of the transformation of solar energy into desalinated water is the main problem that this technology has. Although it is respectful with the environment, its efficiency should always be compared with other existing methods.
Comments
The title of the article does not correspond to its content. An important transformation, both modeling and experimental, should be carried out for the article to be finalized.
Only the water heating part has been worked on, and there is no content whatsoever on the additional elements available in the installation: Evaporator, exchanger, auxiliary elements (flow pumps, pressure settings, etc.). An important part of modeling and experimental is missing on the additional elements.
(111) The figure shows the most representative equipment of the installation. What are the auxiliary equipment required (flow variation, sub pressure, control, filters, etc.)
(242) the equation reflects the temporal and spatial variation (t, x). Why has only the permanent regime been considered?
(410) The established theoretical two-phase flow model has been tested for accuracy by varying the inlet flow to the heater. The precision of the system is observed against three setpoints. However, nowhere in the document (theoretical or experimental) the existing temporal study is shown when the flow values ​​are changed at the input.
(468) it is observed that the operating range in a two-phase situation is small and although an improvement in performance is observed, the performance of the system as a whole (heating salted water) has not been quantified.
(520) In the discussion section, the comparative study between the proposed system and the systems already established for water desalination is not exposed.
(520) Neither has the reality of the proposed system been exposed in this work: investment, space required, energy required per cubic meter, capacity of the facilities, scale of the project, etc ...
Round 2
Reviewer 2 Report
The authors have addressed most of my concerns. I believe the manuscript is acceptable for publication in water.
Reviewer 3 Report
I don't have to make any more comments.
This manuscript is a resubmission of an earlier submission. The following is a list of the peer review reports and author responses from that submission.
Round 1
Reviewer 1 Report
The presented article on desalination system design contains numerous factual errors and ambiguities, which is why I do not recommend it for publication in its current form. Below is a list of the most important comments.
Critical remarks:
1. The literature review is poor and not up-to-date. Some of the citations are local publications not indexed by global databases (therefore not available to the reviewer), no recent scientific research (the latest citation from 2016) on desalination processes using solar energy.
2. L. 102 - what is the pressure in the evaporation chamber? Total vacuum or vacuum of a certain degree? Enter the exact value.
3. L. 150 - there is a statement in the paper: 'The solar radiation in Xinjiang is strong'. What does in mean? It would be worth showing the annual solar radiation profile for the installation location.
4. Only general views of the installation are presented in the paper, and the topic clearly specifies the design of the installation. There are no technical drawings showing, for example, heat exchangers or the atomizer described in 176-183, which would facilitate the analysis of the system and allow for a technical review of the project.
5. Fig. 8 - The pipes shown in the picture are not coaxial as the text suggests, but parallel to each other.
6. In the chapter on heat transfer modeling, there is no clear specification of the equations allowing to determine the key parameter, which is the heat transfer coefficient. The authors cite [28], which is unclear for the reviewer (there is no author named Shah).
6. L. 442 - are the selected parameters consistent with the actual operating values ​​of the installation?
7. In my opinion, the work lacks experimental research and analysis of the obtained results, although the actual system is presented. Sections 3.1 and 3.2 relate to numerical research; there is no validation of the numerical model against the actual measurement data.